# Caffeine-Induced Effects on Human Skeletal Muscle Contraction Time and Maximal Displacement Measured by Tensiomyography

**DOI:** 10.3390/nu13030815

**Published:** 2021-03-02

**Authors:** Przemysław Domaszewski, Paweł Pakosz, Mariusz Konieczny, Dawid Bączkowicz, Ewa Sadowska-Krępa

**Affiliations:** 1Faculty of Physical Education and Physiotherapy, Opole University of Technology, 45-758 Opole, Poland; m.konieczny@po.edu.pl (M.K.); d.baczkowicz@po.edu.pl (D.B.); 2Institute of Sport Sciences, The Jerzy Kukuczka Academy of Physical Education, 40-065 Katowice, Poland; e.sadowska-krepa@awf.katowice.pl

**Keywords:** caffeine, TMG, contraction time, professional athletes

## Abstract

Studies on muscle activation time in sport after caffeine supplementation confirmed the effectiveness of caffeine. The novel approach was to determine whether a dose of 9 mg/kg/ body mass (b.m.) of caffeine affects the changes of contraction time and the displacement of electrically stimulated muscle (gastrocnemius medialis) in professional athletes who regularly consume products rich in caffeine and do not comply with the caffeine discontinuation period requirements. The study included 40 professional male handball players (age = 23.13 ± 3.51, b.m. = 93.51 ± 15.70 kg, height 191 ± 7.72, BMI = 25.89 ± 3.10). The analysis showed that in the experimental group the values of examined parameters were significantly reduced (*p* ≤ 0.001) (contraction time: before = 20.60 ± 2.58 ms/ after = 18.43 ± 3.05 ms; maximal displacement: before = 2.32 ± 0.80 mm/after = 1.69 ± 0.51 mm). No significant changes were found in the placebo group. The main achievement of this research was to demonstrate that caffeine at a dose of 9 mg/kg in professional athletes who regularly consume products rich in caffeine has a direct positive effect on the mechanical activity of skeletal muscle stimulated by an electric pulse.

## 1. Introduction

Caffeine (1,3,7-trimethylxanthine) is the most frequently consumed stimulant worldwide. Its capacity to improve exercise performance [1] as well as cognitive functions make it a very common dietary supplement in sports nutrition [2]. The half-life of caffeine is estimated at four to six hours [3]. The wide distribution of caffeine in the body makes it difficult to precisely determine the individual effects of its activity and its impact on sporting performance [4]. It is believed that one of the most important mechanisms of caffeine activity is the direct, antagonistic effect on adenosine receptors [5,6]. This mechanism is based on the prevention of adenosine-induced dopamine release suppression [7,8], which contributes to the stimulation of the body and increased alertness [9]. Preventing the decrease of neuronal activity by blocking the adenosine receptors is associated with the possibility of increasing muscle fiber recruitment [10]. There is also evidence that caffeine affects phosphodiesterase inhibition, resulting in an increased concentration of cyclic adenosine monophosphate (cAMP), increased catecholamine secretion [8], and inhibition of the γ-aminobutyric acid (GABA) receptors [11].

Another mechanism of the caffeine effect is the opening of an ion channel in the sarcoplasmic reticulum. Two types of ion channels responsible for activation of calcium release from intracellular Ca^2+^ stores are known, i.e., Ip3Rs and RyRs. Caffeine has the ability to open the RyRs channel, especially in muscles and myocytes [12,13]. There is a reserve of Ca^2+^ in the sarcoplasmic reticulum (SR), which can be additionally released in the presence of caffeine, resulting in improved muscle speed and strength [14,15]. As a result of the binding to RyRs, the Ca^2+^ reserve is activated, and Ca^2+^ is released into the intracellular space. The myofibrillar sensitivity to calcium ions becomes increased, slowing the calcium pump and improving the SR Ca^2+^ permeability [16]. However, the mobilization of intracellular calcium requires much higher caffeine concentration [17,18] than suppression of the adenosine receptors [19]. Along with the increased mobilization of calcium ions under the influence of caffeine, slowing the rate of muscle relaxation (which is associated with a decrease in the SR Ca^2+^ pump activity) was also observed [16]. Moreover, the muscles exposed to caffeine showed less ability to restore homeostasis. These changes, however, may be associated with increased damage caused by caffeine-induced high muscle work intensity. In experiments on the isolated sarcoplasmic reticulum, the application of caffeine resulted in the immediate release of Ca^2+^ in 11 out of 12 samples, which was associated with caffeine-induced increased release of Ca^2+^, leading to a greater ability of muscles to work under electrical stimulation [20,21]. However, the practical implementation of some study results in sport was hampered by the fact that a large group of in vitro studies was based on high, mmol/l caffeine doses [16,22,23] which demonstrated toxic effects on the human body [5].

James et al. (2004) [24] suggested that the physiological concentration of caffeine achievable with safe doses (3–6 mg/kg are considered safe [25]) was too low to significantly change the concentration of calcium ions, thus affecting the strength and speed of reaction. More recent studies seem to contradict the suggestions of James et al. and prove the direct, ergogenic effect of a physiological dose of caffeine of 70 μmol/l of a nearly 3% increase in acute muscle power output in fast-twitch and 6% in slow-twitch muscle fibers [23,26]. The most commonly used doses of caffeine in human studies ranging from 3 to 9 mg/kg, result in the plasma concentrations varying between 20 and 70 μmol/l [27,28,29,30]. In studies determining the dose-dependent reaction to caffeine [31,32,33], a certain optimal level of caffeine up to a maximum of 9 mg/kg was indicated, above which there is no improvement, or even a possible reduction, in exercise performance [34,35]. Recent in vitro studies also show that the physiological concentration of caffeine can directly affect the muscle fibers by significantly increasing their force, power, and reaction rate [36]. 

The ability of caffeine to boost adrenaline rush, release calcium ions, improve Na⁺/K⁺-ATPase and reduce pain perception [37] seems to be directly related to improved sports performance [38]. Nevertheless, it is unclear whether the improvement in the rate of muscle contraction after the use of caffeine depends mainly on its effect on the central nervous system (CNS), or whether the mechanism of improvement in the rate of contraction is mainly due to the effect of caffeine on the molecular level of the muscle cell. In vitro studies have repeatedly demonstrated that caffeine can increase muscular fibers excitability induced by a single electrical stimulus, but researchers disagree whether the physiological level of caffeine reached after supplementation is sufficient to induce significant changes in the rate of muscle fiber contraction in vivo [39]. 

In the present study, tensiomyography (TMG) was used, in which an electrical pulse induces muscle contraction independent of the central nervous system [40]. TMG uses a high-precision (4 mm) digital transducer placed perpendicularly to the muscle surface, capable of assessing different parameters extracted from its waveform after a submaximal-to-maximal percutaneous neuromuscular stimulation [41]. Electrical stimulation is delivered with two surface electrodes placed proximal and distal to the sensor tip. Displacement–time curve recordings allow muscle contractile properties to be assessed, which include maximal radial displacement (Dm), contraction time (Tc), delay time (Td), sustain time (Ts), and half-relaxation time (Tr). Between these five TMG parameters, Dm and Tc are generally considered the most valid [42]. Dm presents an excellent relative reliability (ICC 0.94–0.95) but relatively poor absolute reliability (CV 12.30–9.71%). Tc, on the other hand, shows only moderate to good ICCs with wide confidence intervals (0.45–0.95) but acceptable levels of absolute reliability (CV below 8%) [43]. These parameters provid a theoretical assessment of muscle fiber status, i.e., shorter Tc of biceps femoris have been associated with faster running speed [44]. TMG assesses contractions of superficial muscles without any effort from the examined person. It ensures gaining fast and accurate information with no significant involvement and interference of the athletic training process [45,46]. 

Considering the aforementioned results, the main goal of this study was to assess the acute effect of 9 mg/kg dose of caffeine on the mechanical activity of skeletal muscle stimulated by an electric pulse. We hypothesized that doses of 9 mg/kg of anhydrous caffeine would change muscle activation time in professional athletes who regularly consume products rich in caffeine and do not comply with the caffeine discontinuation period requirements.

## 2. Materials and Methods

Forty professional male handball players (third and second team in Poland league in 2019) (age = 23.13 ± 3.51, b.m. = 93.51 ± 15.70 kg, BMI = 25.89 ± 3.10) were included in the study. The athletes were classified as high habitual caffeine consumers (351 ± 139 mg/d) [27] based on the Food Frequency Questionnaire (FFQ). The participants were asked not to change their caffeine consumption habits before the day of experiment. Because some researchers suggested that caffeine was more effective in the morning [47,48], all measurements were taken between 8.00 a.m. and 10.30 a.m., at least two hours after a light breakfast. On the day of the study, the players were asked to refrain from products containing caffeine. The exact body composition was determined using the SECA mBCA 515 analyzer (Seca GmbH&Co. KG, Hamburg, Germany). The study used a randomized, double-blind, placebo control design. The players were randomly divided into two groups: placebo (control) (CON, *n* = 20,) and experimental (with caffeine supplementation) at a dose of 9 mg/kg/b.m. (EXP, *n* = 20). The calculated to the body mass caffeine (ALLNUTRITION, caffeine 200 power) dose was administered 60 min before the TMG measurement in the transparent cellulose capsules (hydroxypropyl methylcellulose 100%). Players from the placebo group received identical capsules filled with pure potato starch. The capsules were washed down with spring water. The examiners conducting the TMG test were not aware of which group they were performing the test on in order not to consciously influence the determination of TMG parameters.

### 2.1. Subjects

The characteristics of the participants are presented in Table 1. The exclusion criteria included individuals after fresh trauma and injuries, and those reporting high sensitivity to caffeine, manifested by sleeplessness, headaches, anxiety, irritability, or tremor. The aims of the study were approved by the Bioethics Committee of the Regional Medical Chamber in Opole No. 260, Poland, in accordance with the Declaration of Helsinki guidelines concerning research on human subjects. All participants were informed of the benefits and risks of the investigation prior to signing an institutionally approved informed consent document to participate in the study. 

### 2.2. Procedures 

The gastrocnemius medialis muscle of the right leg (all the handball players were right-footed) was chosen for the TMG measurement, because it showed a very low measurement error [49]. Two muscle parameters were selected for the analysis: Tc—contraction time (ms), i.e., muscle response time, and Dm—maximal displacement (mm). Decreased Tc values indicate a higher fast-twitch muscle fiber proportion. Decreased Dm value indicates a higher muscle tone [50]. The participants were lying in a prone position on the examination couch. The correct angle in the joints allowing for relaxation of the examined muscles was ensured using a semi-roller placed under the ankle joints. The examination was carried out in accordance with the guidelines and recommendations of the device manufacturers (GK 40 Panoptik d.o.o., Ljubljana, Slovenia). Two self-adhesive electrodes (Axelgaard, Pulse) stimulating the muscle were placed 2–5 cm apart. The electrodes were placed in a way that did not affect the tendons and allowed isolating the contraction of the particular muscle and avoiding simultaneous activation of nearby muscles. The placement of the sensor was selected manually in order to locate the thickest and most deformable part of the muscle. If necessary, the placement of the sensor was adjusted during the test to obtain the best mechanical response of the muscle. The sensor was applied to the skin halfway between the electrodes. 

The electrodes received one 1-ms single-phase rectangular pulse from the electro-stimulator (TMG-S1, Furlan & Co. ltd., Ljubljana, Slovenia) inducing percutaneous muscle contraction. The pulse power was gradually increased by 10 mA until the maximal contraction reaction was achieved. In order to minimize the effects of fatigue, 10-s intervals were taken between the pulse. Typical maximum contraction reactions were recorded between 40 and 80 mA. The TMG digital signal was directly received from the Matlab Compiler Toolbox, using a 1 kHz sampling frequency. The TMG signal was saved and stored on a portable PC. The maximum muscle response was recorded for future analysis. After the first muscle test, the location of the displacement sensor was marked on the body of the examined person so that the second measurement would be performed following the same pattern.

### 2.3. Statistical Analysis

All dependent variables were subjected to two groups (EXP and CON) × 2 times (pre and post) mixed ANOVA with repeated measures of the last factor. Due to the non-normal distributions, log10 transformation was applied. Tukey’s HSD follow-up analyses were applied as post hoc tests. The effect size was calculated as partial eta-squared η^2^p. Effect size: small—d = 0.2; medium—d = 0.5, and large—d = 0.8, alpha value ≤ 0.05 according to Cohen (1988). The collected data were analyzed statistically using the Jamovi 1.6.9. software package (https://www.jamovi.org/download.html accessed on 26 February 2021). 

## 3. Results

Statistical analysis has shown that in both parameters of the experimental group the TMG values decreased significantly (*p* ≤ 0.001), whereas in the control group the results did not differ significantly in both parameters (Table 2). In the experimental group, the mean Tc value decreased significantly (before = 20.60 ms ± 2.58; after = 18.43 ms ± 3.05, and effect size d = 0.77), which indicated a reduction in muscle contraction time after the applied dose of caffeine. The mean Dm value also decreased (before = 2.32 mm ± 0.80; after = 1.69 mm ± 0.51, and effect size d = 0.78), indicating a significant decrease in muscle contraction displacement. In the control group, the mean Tc values were: Before = 20.00 ms ± 3.74 and After = 19.87 ms ± 3.06 and effect size d = 0.07. In the same group, the mean Dm values were: before = 2.06 mm ± 0.66 and after = 1.97 mm ± 0.60, and effect size d = 0.41.

Table 3 shows the ANOVA results of the analyzed Tc parameter. The interaction of the variables Time * Group indicated statistically significant differences (F = 5.27 *p* = 0.022) and effect size η^2^p = 0.131). The post hoc analysis showed significant differences in the experimental group (t = 3.499, *p* = 0.006), which suggests the influence of supplementation on the value of Tc parameters (Table 4). 

The analysis of the Dm parameter (Table 5) showed similar relationships because there was also a statistically significant interaction of the variables Time * Group (F = 14.3 *p* ≤ 0.001) and effect size η^2^p = 0.274. The post hoc analysis showed significant differences in the experimental group (t = 6.040, *p* ≤ 0.001), which suggested the influence of supplementation on the value of Tc parameters (Table 6).

## 4. Discussion

The main finding of the study was that acute caffeine consumption affected both the time of contraction and the maximal displacement.

The review of the latest literature confirms the ergogenic effect of caffeine on aerobic endurance, muscle strength, muscle endurance, power, jumping performance, and exercise speed [1]. In vitro experiments on isolated muscle fibers conducted by Tallis et al. also clearly demonstrate that caffeine can significantly improve the speed and force of the contractions induced by electric impulses [23,26]. In vivo studies on caffeine-induced enhancement confirmed caffeine effectiveness, but did not indicate a reduction in reaction time associated with caffeine-induced changes directly in muscle fibers. The results of the present study verified that caffeine at a dose of 9 mg/kg separately from inducing changes in the central nervous system, also had a direct effect on the mechanical activity of skeletal muscles, significantly improving their Tc (*p* < 0.001) and reducing the Dm (*p* < 0.001). This result may indicate a shortening of movement speed. However, TMG determines a muscle’s response to an external stimulus and not to a stimulus directly from the brain [14,15]. The local effects of caffeine would, to some extent, challenge the notion that the ergogenic effects are solely due to CNS alterations.

Intramuscular changes caused by caffeine, however, slow down the time needed for muscle relaxation. It is believed that reduced activity of the SR Ca^2+^ pump is the underlying mechanism of increased muscle stiffness after caffeine [16]. This mechanism is probably also responsible for the significant reduction in maximal displacement, one of the values considered to be strongly correlated with passive muscle stiffness [43,51]. 

Despite many studies, there are still no clear recommendations for determining the appropriate dose of caffeine according to gender, age, sports, dietary habits, and form of administration [52,53,54]. Researchers usually asked participants to stop using caffeine products before experiment [55], which can significantly affect their sensitivity to caffeine. Recently, Wilk et al. suggest also that even high doses of caffeine were ineffective in high-caffeine consumers, although the dosage of caffeine used pre-exercise was well-above their daily intake of this substance [56]. A widespread practical recommendation for athletes is to remove/reduce caffeine before a sports competition [57]. However, almost 75–90% of professional athletes consume caffeine before or during every training session or sports event [2], and despite recommendations to discontinue caffeine, most often do not follow these guidelines [30,58]. Significant changes in the muscle activation observed in this study suggest that positive effects are achievable without changing the caffeine consumption habits.

In this report, we present experimental data to support a new look at reports of caffeine-induced effects on the muscle reaction rate [4,8,23,26,31,36,59,60,61]. Presumably, the improvement in the rate of electrically stimulated and caffeine-induced muscle contraction is based much more on intramuscular mechanisms regulating the release of calcium ions. 

According to the author’s knowledge, this is the first attempt to integrate and investigate the effects of caffeine on electrically stimulated muscle in vivo; therefore, it is difficult to compare our results to the literature. Future studies implementing caffeine wh electrical muscle stimulation are needed.

## 5. Conclusions

The single dose of 9 mg/kg/b.m. of anhydrous caffeine used in the present study seems to be suitable for achieving a significant effect on the mechanical activity of skeletal muscles, meaningfully improving their contraction time and reducing the maximal displacement in a group of professional athletes, classified as high habitual caffeine consumers who do not comply with the caffeine discontinuation period requirements. 

## 6. Limitations

To provide concrete conclusions in future studies, the authors plan to include a group of participants who are heavy users and stopped consuming caffeine several days before the study.

The authors did not explore the side effects of high doses of caffeine ingestion.

## Figures and Tables

**Table 1 nutrients-13-00815-t001:** Characteristics of study groups.

	Age	BMI (kg/m^2^)	Relative Fat Mass (%)	Weight (kg)	Height (cm)
EXP (*n* = 20)	23.3 ± 3.74	25.59 ± 3.74	18.89 ± 6.41	93.12 ± 17.51	190 ± 8.44
CON (*n* = 20)	22.95 ± 3.35	26.19 ± 2.71	17.79 ± 9.43	98.02 ± 14.11	189 ± 6.84

BMI–body mass index; EXP–experimental group; CON–control group.

**Table 2 nutrients-13-00815-t002:** Descriptive statistics of the parameters Tc—Contraction Time (ms), and Dm—Maximal Displacement in both groups.

	X ± SD	P
EXP		
Tc (ms) Before	20.60 ± 2.58	0.001
Tc (ms) After	18.43 ± 3.05	
Dm (mm) Before	2.32 ± 0.80	0.001
Dm (mm) After	1.69 ± 0.51	
CON		
Tc (ms) Before	20.00 ± 3.74	0.648
Tc (ms) After	19.87 ± 3.06	
Dm (mm) Before	2.06 ± 0.66	0.070
Dm (mm) After	1.97 ± 0.60	

**Table 3 nutrients-13-00815-t003:** Analysis of the Tc parameter within subjects effects.

	Sum of Squares	Df	Mean Square	F	P	η^2^p
Time	0.0137	1	0.01374	6.53	0.015	0.147
T*G	0.0120	1	0.01203	5.72	0.022	0.131
Residual	0.0799	38	0.00210			

T*G–interaction between time and group.

**Table 4 nutrients-13-00815-t004:** Analysis of the Tc parameter post hoc comparisons-Time * Group.

Comparison								
Time	Group	Time	Group	Mean Difference	SE	Df	T	P_tukey_
Pre	EXP	Pre	CON	0.01563	0.0220	57.5	0.709	0.893
		Post	EXP	0.05074	0.0145	38.0	3.499	0.006
		Post	CON	0.01731	0.0220	57.5	0.786	0.861
	CON	Post	EXP	0.03511	0.0220	57.5	1.594	0.390
		Post	CON	0.00168	0.0145	38.0	0.116	0.999
Post	EXP	Post	CON	−0.03343	0.0220	57.5	−1.517	0.434

**Table 5 nutrients-13-00815-t005:** Analysis of the Dm parameter within subjects effects.

	Sum of Squares	Df	Mean Square	F	P	η^2^p
Time	0.1191	1	0.11910	22.6	<0.001	0.373
T*G	0.0753	1	0.07533	14.3	<0.001	0.274
Residual	0.2000	38	0.00526			

T*G–interaction between time and group.

**Table 6 nutrients-13-00815-t006:** Analysis of the Dm parameter post-hoc comparisons—Time * Group.

Comparison								
Time	Group	Time	Group	Mean Difference	SE	Df	T	P_tukey_
Pre	EXP	Pre	CON	0.0584	0.0458	48.7	1.273	0.584
		Post	EXP	0.1385	0.0229	38.0	6.040	< 0.001
		Post	CON	0.0742	0.0458	48.7	1.618	0.378
	CON	Post	EXP	0.0802	0.0458	48.7	1.749	0.310
		Post	CON	0.0158	0.0229	38.0	0.689	0.901
Post	EXP	Post	CON	−0.0644	0.0458	48.7	−1.405	0.503

## Data Availability

The datasets generated and/or analyzed during the current study are not publicly available. However, the data are available from the corresponding author on reasonable request.

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
