# Peer review of "Caffeine-Induced Effects on Human Skeletal Muscle Contraction Time and Maximal Displacement Measured by Tensiomyography"

_nutrients, 2021, doi:10.3390/nu13030815_

Round 1

Reviewer 1 Report

This is an interesting study, and I only have a few minor comments for the authors to consider.

Line 36-38 – this depends on the source of caffeine. I believe that it would be best to revise this sentence or remove it completely.

Line 59 – could the authors reference here the meta-analyses that demonstrated an ergogenic effect of caffeine on strength and speed? Please see the work by Warren et al. and Grgic et al.

Line 88 – is this correct? Would this dose be considered the maximum physiological dose observed in humans? If so, it would be associated with major side-effects. Could the authors please re-check these data?

Line 94 – a dose of 12 mg/kg is definitely not needed for an ergogenic effect. Could the authors please make this point clear to prevent misunderstanding among readers?

Line 121 – 124 – here, it would be of benefit to briefly describe the physiological relevance of these variables. For example, what data do we get for the half-relaxation time, and what is its relevance to the practitioner?

Line 149 – why did the authors use 9 mg/kg? What was the rationale for this approach?

Line 176 – again, here, it would be of benefit to describe the data that we get from these two variables and their physiological importance (from a sporting or health perspective).

It would be highly relevant to add a section that would focus on the reliability of these outcomes. What is the ICC and/or CV for test-retest reliability for these variables?

Line 212 – add a scale for effect size interpretation and the alpha value set for the study.

Additionally, using Cohen's d for effect size interpretation would likely much more intuitive to interpret for exercise practitioners.

The data in table 4 is oddly presented and difficult to interpret. Why is there only one pre-condition and five post-conditions? Make sure to revise the tables (where needed) to ensure that the data is easy to interpret. Same for table 6. Either revise these tables or add more detailed explanations in the table descriptions.

Line 274 – this sentence does not read well; please revise.

Line 300 – as the authors point out, it is commonly acknowledged that the ergogenic effects of caffeine on exercise performance are due to caffeine's effects on adenosine receptors. However, given that your study showed local effects of caffeine (i.e., gastrocnemius medialis muscle), this would, to a certain extent, challenge the notion that the ergogenic effects are solely due to CNS alterations. Could the authors please incorporate these points as speculation in the discussion?

Additionally, given that the authors observed a decrease in contraction times, this would suggest that caffeine ingestion has a great effect on movement velocity. Indeed, this recent meta-analysis (and several other primary studies) have demonstrated a decrease in mean and peak velocity in resistance exercise following caffeine ingestion. Given that the authors observed a decrease in contraction time, this could be linked to the changes in movement velocity previouslly observed.

Meta-analysis: https://link.springer.com/article/10.1007/s40279-019-01211-9

Line 301-316 – these are some solid points but please also add in that a greater effect of caffeine might have been observed if the athletes stopped caffeine intake and that future studies may consider exploring this further. Without actually including a group of participants that are high users and that stoped caffeine intake a few days prior to testing, we cannot provide any definite conclusions.

Reviewer 2 Report

Dear Editor,

Many thanks for the opportunity to review the manuscript titled “Caffeine-induced effects on human skeletal muscle contraction time and maximal displacement measured by tensiomyography”, the authors have conducted a novel study investigating the effects of caffeine ingestion on human skeletal properties. However, there are some major issues that need to be addressed if the authors would like to publish their investigation in a prestigious journal such as Nutrients.

ABSTRACT

Line 14: Authors in this study measured reaction time or muscle contration time?, according to this line is unclear. Please clarify.

Line 16. Why the authors selected 9mg/kg and stated that is novel with this dose? Previous studies have analyzed the effects of caffeine and TMG? In addition, delete b.m repetitive along the manuscript..

Line 21: Add heigth data of handball players

INTRODUCTION

The Introduction is very extended although should explore more deeply the relationship between caffeine intake and muscle contractile properties and more important what is the reason for studying the ingestion of high doses of caffeine instead low doses of caffeine, due to such as previously mentioned high doses of caffeine have been associated with higher side effects. Please explain carefully this point.

Line 37: Once ingested, after only 30 minutes caffeine effects sports performance? Please add a reference or change your statement.

Line 81: Safe doses? Please clarify.

Line 90-95: Rephrase this paragraph, due to some of your stataments could be incorrect. Are the authors are completely sure that are neccesary 12 mg/kg/ bm for obtaining exercise benefits? In addition, the authors considered the seconday effects associated to higher doses. (doi: 10.1249/MSS.0b013e31829a6672).

METHODS

The authors should add more information about the experimental design is a bit unclear for the potential readers. There is one major limitation of this study that needs to be clearly acknowledged. From what it is presented, it seems that the authors did not explore the effectiveness of the blinding. This is important as supplement identification appeared to influence exercise outcomes and could be a source of bias in sports nutrition. If the authors did not explore this, it should be clearly stated as a limitation. See these papers for a discussion on this topic

https://www.ncbi.nlm.nih.gov/pubmed/27882605

https://link.springer.com/article/10.1007/s40279-018-0997-y

Line 136: Please add more information about the “professional leve of your study participants” as per guidance (https://www.bases.org.uk/imgs/51_article_p6527.pdf).

Line 149: Please add caffeine and placebo details (brand, etc).

Line 170= Some mistakes in table nº1 in age of control group (no SD data), relative fat mass, etc.

Line 172: All the measurements are realized in the same time of day? Due to caffeine is influenced by time-of-day when it is ingested. (doi: 10.1371/journal.pone.0033807, 10.1016/j.jsams.2014.04.010).

Line 192: Why the authors don´t explore the side effects associated to high doses of caffeine ingestion in handball players?, Specially when previous studies in handball players have analyzed this side effects (doi: 10.1123/ijspp.2019-0847).

RESULTS

All the tables are completely necessary?,please explore the possibility of eliminate some manuscript tables?

DISCUSSION

The discussion is weak and more information is neeeded, I encourage to the authors for working more deeply in this section. In addition, a limitations paragraph should be added.

REFERENCES

Some mistakes in references list

Reference 9 (without article pages).

Reference 38: No journal name,

Round 2

Reviewer 2 Report

Dear authors, 

The manuscript have improved considerably, however some point need to be elucidated before of being considered of acceptance in Nutrients Journal:

INTRODUCTION
Line 94-97: In studies determining the dose-dependent reaction to caffeine, a certain optimal level of caffeine up to a maximum of 12 mg/kg/b.m. was indicated, above which there is no improvement, or even a possible reduction, in exercise performance. This statement is incorrect, please see the following studies and correct your statement in the manuscript version (doi: 10.1152/jappl.1995.78.3.867; doi: 10.1249/MSS.0b013e31829a6672).

METHODS

Line 149: The handball players belongs to the same handball team? Please clarify.

Line 154-155: Please add a reference that support your statement (doi: 10.1371/journal.pone.0033807, 10.1016/j.jsams.2014.04.010).

Line 165: Please add country.

Line 200: The authors stated "Decreased Tc values indicate a higher fast-twitch muscle fiber proportion." Could the authors add more information or add a reference that support their statement?

RESULTS:

Line 237-238: The authors calculated the effect sizes according to Cohen et al (1988)?. Please clarify

DISCUSSION:

Line 317-319: Please rewording.

REFERENCES:

Reference nº 31 Page names are incorrect, please correct:

Reference nº 43: Some authors name are missing in this reference. Please correct.

Reference nº53: Please change "C Onsiderations". by "considerations"
